

# Foot pressure distribution in White Rhinoceroses (*Ceratotherium simum*) during walking

Olga Panagiotopoulou[1], Todd C. Pataky[2] and John R. Hutchinson[3]

[1] Monash Biomedicine Discovery Institute, Department of Anatomy and Developmental Biology, Moving Morphology & Functional Mechanics Laboratory, Monash University, Clayton, VIC, Australia
[2] Department of Human Health Sciences, Kyoto University, Kyoto, Japan
[3] Department of Comparative Biomedical Sciences, Structure and Motion Laboratory, Royal Veterinary College, Hatfield, UK

Corresponding authors
Olga Panagiotopoulou,
olga.panagiotopoulou@monash.edu
John R. Hutchinson,
JHutchinson@rvc.ac.uk

## ABSTRACT

White rhinoceroses (*Ceratotherium simum*) are odd-toed ungulates that belong to the group Perissodactyla. Being second only to elephants in terms of large body mass amongst extant tetrapods, rhinoceroses make fascinating subjects for the study of how large land animals support and move themselves. Rhinoceroses often are kept in captivity for protection from ivory poachers and for educational/touristic purposes, yet a detrimental side effect of captivity can be foot disease (i.e., enthesopathies and osteoarthritis around the phalanges). Foot diseases in large mammals are multifactorial, but locomotor biomechanics (e.g., pressures routinely experienced by the feet) surely can be a contributing factor. However, due to a lack of in vivo experimental data on rhinoceros foot pressures, our knowledge of locomotor performance and its links to foot disease is limited. The overall aim of this study was to characterize peak pressures and center of pressure trajectories in white rhinoceroses during walking. We asked two major questions. First, are peak locomotor pressures the lowest around the fat pad and its lobes (as in the case of elephants)? Second, are peak locomotor pressures concentrated around the areas with the highest reported incidence of pathologies? Our results show a reduction of pressures around the fat pad and its lobes, which is potentially due to the material properties of the fat pad or a tendency to avoid or limit "heel" contact at impact. We also found an even and gradual concentration of foot pressures across all digits, which may be a by-product of the more horizontal foot roll-off during the stance phase. While our exploratory, descriptive sample precluded hypothesis testing, our study provides important new data on rhinoceros locomotion for future studies to build on, and thus impetus for improved implementation in the care of captive/managed rhinoceroses.

## INTRODUCTION

Over millions of years of evolution, the feet of rhinoceroses have had to change with other alterations of limb morphology, locomotor behavior, body size, habitat, and more

(*Prothero, 2005*; *Stilson, Hopkins & Davis, 2016*). Extant rhinoceroses include the second largest (after elephants) terrestrial mammals, with body masses in the White rhinoceros reaching up to 3,600 kg (*Groves, 1972*; *Hillman-Smith et al., 1986*; *Owen-Smith, 1992*). Thus in large rhinoceroses locomotor stresses might be considerable if not well-controlled, imposing severe biomechanical constraints on form and function (*Alexander & Pond, 1992*). Contrary to the feet of elephants, which bear five functional digits and "predigits" (*Hutchinson et al., 2011*; *Mariappa, 1986*; *Neuville, 1935*; *Weissengruber et al., 2006*), rhinoceros feet have three digits (numbered II–IV) terminating in horns/hooves (*Prothero, 2005*; *Regnault et al., 2013*) and no supportive "predigits." Of the three digits, digit II and IV, respectively, dominate the medial and lateral aspects of the foot, whilst digit III is the central and largest of all. Each digit consists of three phalanges (proximal, medial, and distal) and the foot caudally and centrally is enclosed in a fat pad. The bi-lobed fat pad is structurally similar but smaller in size to elephant fat pads and expands when compressed (*Von Houwald, 2001*). This structure potentially helps to evenly distribute locomotor stresses across the sole of the foot, as in the case of elephants (*Panagiotopoulou et al., 2012*, *2016*). Overall, the morphology of rhinoceros feet is fairly symmetrical from medial to lateral, unlike the feet of elephants which are more robust laterally (e.g., digits III–V).

Considering that large mammals' feet support their body mass, understanding healthy foot function is important for understanding healthy gait. This is particularly imperative in view of documented rhinoceros foot pathologies (*Dudley et al., 2015*; *Flach et al., 2003*; *Galateanu et al., 2013*; *Harrison et al., 2011*; *Jacobsen, 2002*; *Jones, 1979*; *Regnault et al., 2013*; *Von Houwald, 2001*; *Von Houwald & Guldenschuh, 2002*; *Von Houwald & Flach, 1998*; *Zainuddin et al., 1990*). Previous research on museum specimens found a high occurrence of enthesopathies and osteoarthritis on the phalanges of rhinoceros feet (*Regnault et al., 2013*)—of the 81 feet from 27 rhinoceroses studied, 54 feet from 22 individuals exhibited osteopathologies (*Dudley et al., 2015*). Surprisingly, limb osteopathologies have remained common in rhinocerotid species across their evolution but increasing with estimated body mass. This is consistent with tradeoffs and compromises between large size, cursorial/mediportal morphology or athletic capacity, and limb health (*Stilson, Hopkins & Davis, 2016*).

Many factors can cause foot disease in large mammals, but previous research in elephants has linked foot disease with obesity, space limitations and the time the animals spent walking or standing on hard (unnatural) surfaces (*Csuti, Sargent & Bechert, 2001*; *Fowler & Mikota, 2006*; *Miller, Hogan & Meehan, 2016*). Our prior studies proposed that elephants normally have high pressures laterally, on digits III–V (*Panagiotopoulou et al., 2012*, *2016*), congruent with where elephants tend to exhibit greater incidences of osteopathologies (*Regnault et al., 2017*). In contrast, there are almost no in vivo studies of locomotion in rhinoceroses (*Alexander & Pond, 1992*), in any aspects including the pressures experienced by the feet. Based on the roughly equivalent occurrence of osteopathologies across rhinoceros digits II–IV (*Regnault et al., 2017*), we expect that pressures would be evenly distributed across these digits too, and for pressures to be low on

the fat pad lobes, without the mediolateral asymmetry of pathologies or pressures observed in elephant feet.

In this pilot study, we describe in vivo locomotor foot pressures and center of pressure trajectories (COP) in three white rhinoceroses (*Ceratotherium simum*) during walking. Our limited sample size does not allow us to conduct hypothesis testing on foot pressure magnitudes. However, we were able to conduct preliminary, qualitative evaluation of our two exploratory hypotheses, for future studies to expand on:

Hypothesis I. Peak locomotor pressures will be the lowest in the central and caudal parts of the foot at the locations of the fat pad and its lobes. This is expected from a dynamic interaction of behavioral walking preferences (manifested in COP) and the compliant properties of the fat pad, as we have previously observed in elephants (*Panagiotopoulou et al., 2012*, *2016*).

Hypothesis II. Peak locomotor pressures will be concentrated equally around the horns/hooves and phalangeal pads of all digits (II–IV), which correspond to the overlying bony areas with the highest evidence of osteoarthritis and similar pathologies (*Regnault et al., 2013*), without a strong tendency for more lateral prevalence of pathology.

## METHODS

### Subjects

Four adults and a juvenile captive southern White Rhinoceros (*Ceratotherium simum*) from Colchester Zoo (Colchester, UK) participated in the study, however, only data from two adults and one juvenile could be used for further analyses (Table 1). The body masses of the subjects were estimated by the zoo keepers using the zoo's records. Zoo keepers and veterinarians gave clinical consent for the study and all animal participants were healthy. The study was approved by The Royal Veterinary College's Animal Ethics Committee (approval number URN 2010 1052).

### Data collection

A five m walkway was constructed in a crush area in the rhinoceros enclosure (Fig. 1A). A three m long and 0.4 m wide foam pad was laid at the beginning of the walkway and was followed by a $1.0 \times 0.4$ m pressure plate (fitted with 8,192 sensors, 2.05 sensors cm$^{-2}$) (Footscan; RSscan, Olen, Belgium), and a one m length of foam pad. The walkway was covered with a 0.3 mm thick rubber mat to prevent the animals from recognising the location of the pressure plate. Reflective tape was placed on the rhinoceros hip and shoulder to calculate walking speeds using a Sony HDR (Sony, London, UK) high definition video camera. The camera was placed perpendicular to and at a two m distance from the walkway. Camera and pressure plate sampling frequencies were respectively 25 and 250 Hz. The pressure plate was calibrated using a known weight (~95 kg human standing on the plate) as per manufacturer's instructions. While we do report absolute pressure magnitudes, the main outcome of interest was the relative (i.e., within-foot) pressure distribution, as this reflects foot functionality. Absolute pressure errors are unexpected to affect relative pressure values. The rhinoceros were guided over the walkway using food

**Table 1 Subject characteristics (*Ceratotherium simum*).** Number of steps refers to the spatially and temporally complete steps used for further analysis in this study. Trials (multiple steps) refer to all trials collected during the in vivo experiments. Steps per foot refer to the individual steps per foot and subject collected during the in vivo experiments.

|  | Subject 1 | Subject 2 | Subject 3 |
|---|---|---|---|
| Age | Adult | Juvenile | Adult |
| Sex | Female | Female | Female |
| Body mass (kg) estimated | 2,500 | 1,000 | 2,500 |
| Shoulder height (m) | 1.5 | 0.65 | 1.42 |
| Mean Froude number | 0.014 | 0.001 | 0.054 |
| Mean velocity (ms$^{-1}$) | 0.46 | 0.60 | 0.87 |
| Mean maximum pressure (N cm$^{-2}$) Fore Left 1 | 23 | 13 | No spatially or temporally complete data |
| Mean maximum pressure (N cm$^{-2}$) Fore Right 2 | 28 | 9 | No spatially or temporally complete data |
| Mean maximum pressure (N cm$^{-2}$) Hind Left 3 | 18 | 4 | 12 |
| Mean maximum pressure (N cm$^{-2}$) Hind Right 4 | 2 | No spatially or temporally complete data | No spatially or temporally complete data |
| Number of steps | 10 | 8 | 5 |
| Trials (multiple steps) | 60 | 38 | 115 |
| Steps, Fore Left | 29 | 17 | 51 |
| Steps, Fore Right | 37 | 12 | 47 |
| Steps, Hind Left | 15 | 9 | 13 |
| Steps, Hind Right | 10 | 13 | 12 |

**Note:**
Due to equipment calibration limitations, absolute pressure values may be inaccurate (see Methods), but relative pressure values across subjects and feet are expected to be reliably quantified.

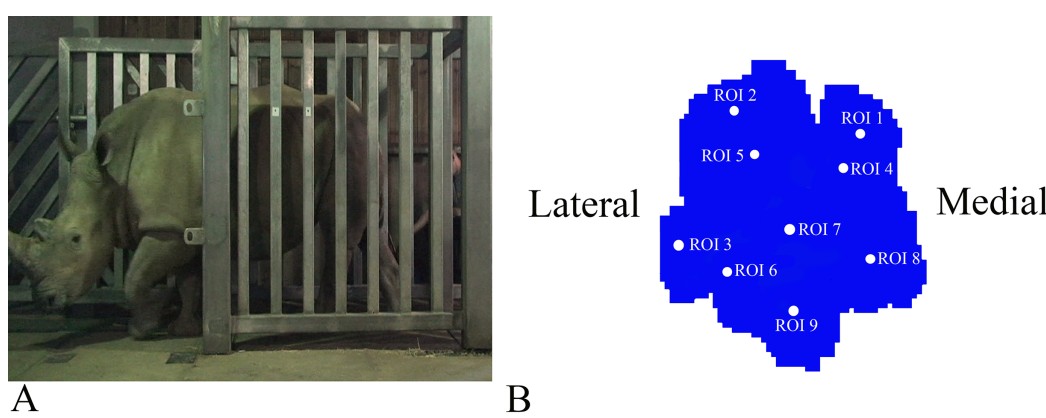

**Figure 1 Schematic illustration of the position of the pressure plates and the nine regions of interest (ROI) during data collection.** (A) Image of Subject 3 walking on the pressure plate in the experimental walkway. (B) Schematic representation of the anatomical location of the nine (ROIs across the left forefoot.)

as encouragement, an average of 20 times each. Trials with obvious acceleration and deceleration (as judged by video) during data collection were excluded from further analysis (*Panagiotopoulou et al., 2012*, *2016*). Animal discomfort was kept to the minimum by stopping data collection when animals appeared disengaged.

## Data processing

Data analysis protocols were similar to *Panagiotopoulou et al. (2012*, *2016)*, implanted in Canopy v. 2.1.8 using SciPy v. 0.19, NumPy 1.11.3 and Matplotlib 2.0 (Enthought Inc., Austin, TX, USA). In brief, the raw pressure data ($x$, $y$, time) of the individual footsteps were exported from the Footscan system, isolated algorithmically using spatio-temporal gaps between clusters of non-zero pressure voxels and were assessed for spatial and temporal completeness as per *Panagiotopoulou et al. (2012*, *2016)*. Individual images representative of spatio-temporally complete footsteps were manually identified as fore/hind, right/left; spatially scaled by a factor of 1.5, using bilinear interpolation to compensate for the non-square measurement grid of the RSscan system (7.62 x 5.08 mm, manufacturer specified); and spatially registered within subjects and feet (see *Panagiotopoulou et al., 2012*). Following scaling and registration, nine anatomically homologous regions of interest (ROIs) were selected on the mean images for each foot as per *Panagiotopoulou et al. (2012*, *2016)*, and peak pressures (N cm$^{-2}$) of the whole stance phase were extracted from a three-pixel radius using a Gaussian kernel mean window with a standard deviation of one pixel. ROIs 1–3, respectively, represented the horns of digits II–IV, ROIs 4–6 represented the (inter) phalangeal pads of digits II–IV, respectively, ROI 7 represented the caudal most ("heel") aspect of the sole and ROIs 8–9 were, respectively, placed on the medial and lateral footpads of the sole (Fig. 1B). COP were computed as the pressure-weighted image centroids' time series after thresholding the images at 0.5 N cm$^{-2}$. Due to the limited number of subjects and steps, the dependent variables were not tested for significance. This was a preliminary, qualitative study of rhinoceros foot function during gait, so we neither derived nor tested a null hypothesis.

## RESULTS

The mean walking speed of all three subjects was 0.53 ms$^{-1}$ (Table 1), which corresponded to a mean Froude number (*Alexander & Jayes, 1983*; Fr = velocity$^2$ ∗ (9.81 ms$^{-2}$ ∗ shoulder height) $^{-1}$) of 0.013, consistent with a slow walk. The peak pressure data per ROI, subject and feet are shown in Table 1. All peak pressure data are in Data S1. Raw pressure data including all trials and steps are available on Figshare (https://doi.org/10.6084/m9. figshare.7608797.v1). The mean peak pressure values for the adult subjects 1 and 3 and all feet were, respectively, 22 N cm$^{-2}$ and 18 N cm$^{-2}$, whilst the mean peak pressure values of the juvenile subject 2 were 0.9 N cm$^{-2}$. The mean peak pressure values for both adult subjects and all feet (20 N cm$^{-2}$) were, respectively, 4.7 and 2.8 times lower than those previously recorded on African (94.6 N cm$^{-2}$) and Asian elephants (56.7 N cm$^{-2}$) during walking (*Panagiotopoulou et al., 2012*, *2016*). The Asian elephant data were collected using the same RSscan system as in this study, yet the African elephant pressure data were collected using a lower resolution system (Zebris Medical GmbH, Biomechanix, Munich)

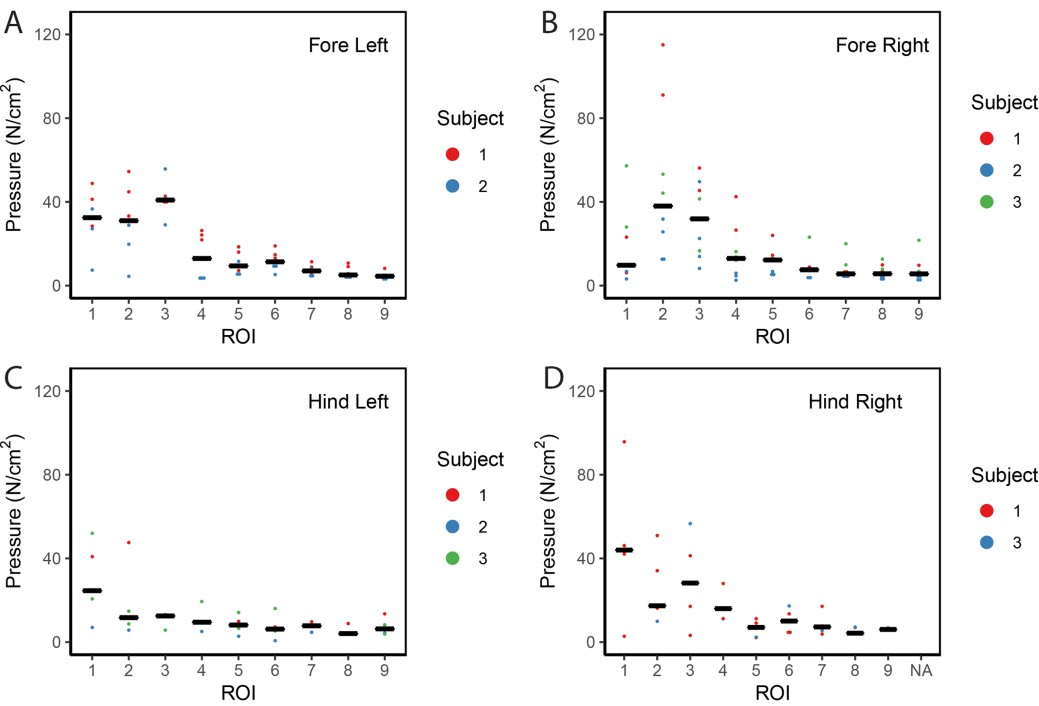

**Figure 2** Scatter plot of peak foot pressure data from all three subjects at the nine regions of interest (ROI) across (A) the left forefoot, (B) right forefoot, (C) left hindfoot, and (D) right hindfoot. Black line represents the median pressure found at each ROI.  

with 100 Hz sampling frequency, sensors resolution of ½ inch and sensor size of 1.27 × 1.27 cm, so the present study's data are not comparable with those prior data. Our data showed that, similar to elephants and other quadrupeds, the forefeet had higher mean pressure magnitudes than the hindfeet for all subjects (Table 1).

Contrary to elephants (*Panagiotopoulou et al., 2012, 2016*), the rhinoceroses' foot pressures did not follow a consistent pattern between feet. The forefeet for adult subject 3 and the juvenile rhinoceros (subject 2) showed higher pressures around the horn of digits II (ROI 1), III (ROI 2), and IV (ROI 3). Intermediate pressures were recorded around the phalangeal pads of digits II–IV and the lowest pressures around the fat pad (ROIs 7–9) (Figs. 2–4). The highest median foot pressures for the right forefeet of all three rhinoceroses were at the horn of digits III and IV, corresponding to ROIs 2 and 3 (Figs. 2 and 4). The lowest median peak pressures were recorded around the fat pad; nevertheless, median peak pressures around the phalangeal pads of all digits were very low. Median pressures for the left hindfeet were the highest for the horn of digit II, followed by ROIs 2 and 3 (Figs. 2 and 5). Intermediate median pressures were recorded at ROIs 4, 5, and 7 and the lowest peak pressures were computed at ROI 8. Regardless, median peak pressure differences between ROIs 2–9 were minimal (Fig. 2; Data S1). Median peak pressures for the hindfeet of the two adult subjects (subject 1 and 3) gave the highest median peak pressures at the horn of digit II (ROI 1) and intermediate pressures at ROIs 2–4 (Figs. 2 and 6). The lowest peak pressures were found at ROIs 5–9.

## Fore Left

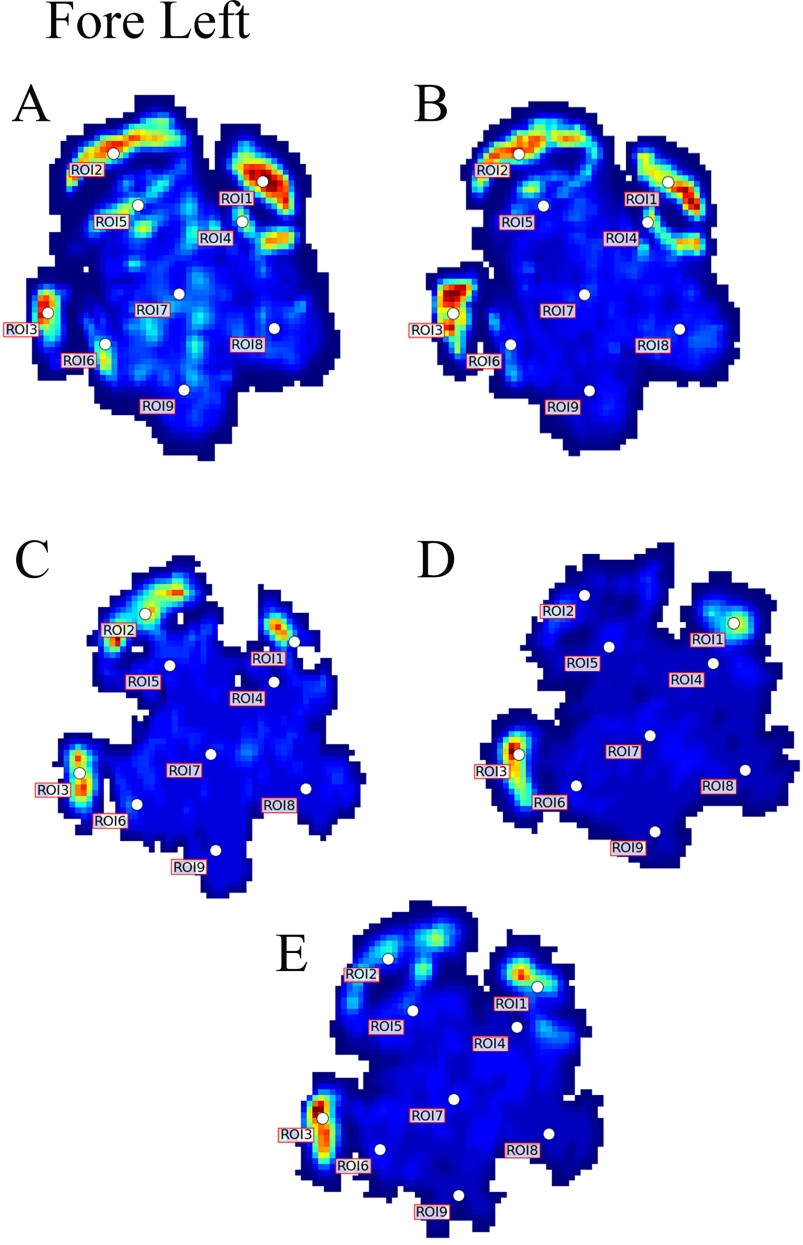

**Figure 3 Peak pressure patterns during the whole stance phase of the left forefoot of subjects 1 (A and B) and 2 (C–E).**

   The COP trajectories for all time frames, animal participants and feet are shown in Figs. S1–S8. Most COP traces began at the medial aspect of the foot caudally to the interphalangeal pad of digit II or at the medial footpad of the sole, then shifted caudally around the heel aspect of the sole and finally passed cranially through digit III by toe-off. Contrary to this caudo-medial and centrally-focused pressure pattern, pressure traces in two trials for the left hindfoot started laterally on digit IV before shifting caudo-cranially and through digit III by toe-off. Thus there was some unusual variability in our subjects' COP traces during normal locomotion.

## Fore Right

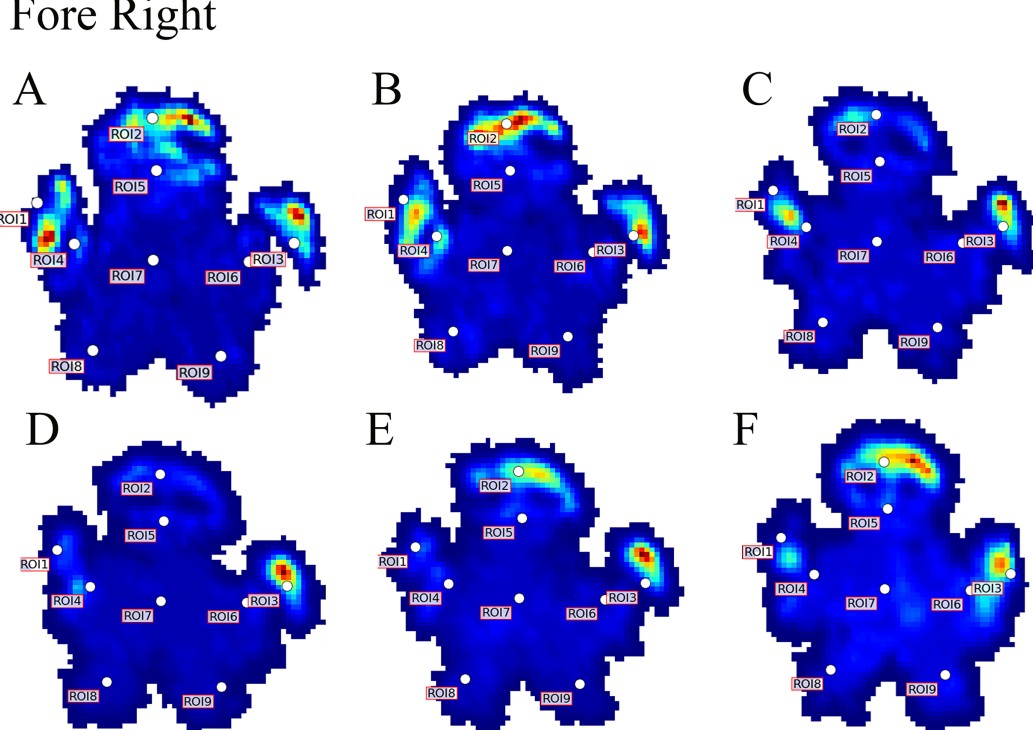

**Figure 4 Peak pressure patterns during the whole stance phase of the right forefoot of subjects 1 (A and B) and 2 (C–F).**

## Hind Left

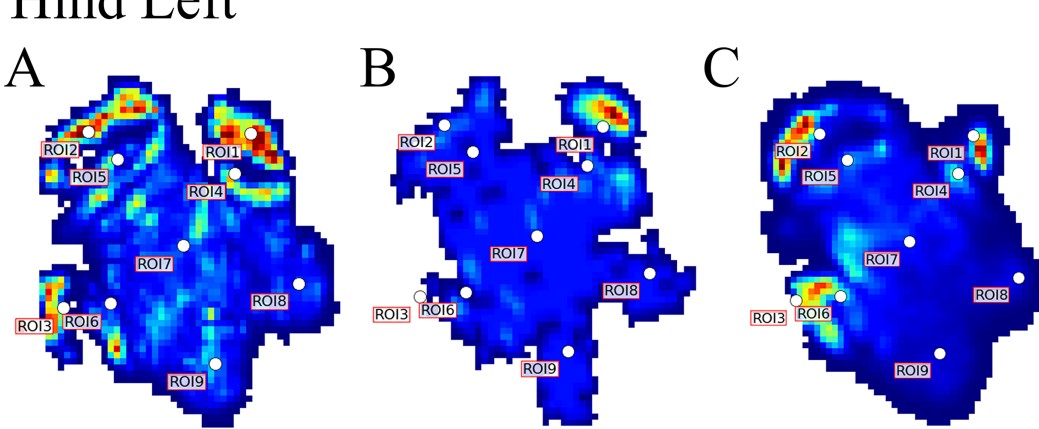

**Figure 5 Peak pressure patterns during the whole stance phase of the left hindfoot of subjects 1 (A), 2 (B), and 3 (C).**

## DISCUSSION

Overall, we found reduction of peak pressures around the fat pads of the feet, qualitatively supporting our hypothesis I that, like in elephants, rhinoceros fat pads may keep locomotor pressures low due to their compliance. Whilst our quantitative results showed variation in peak foot pressures across feet, we recorded the highest peak pressures around the horn and phalangeal pads of all digits, yet this signal was not as strong for the left

# Hind Right

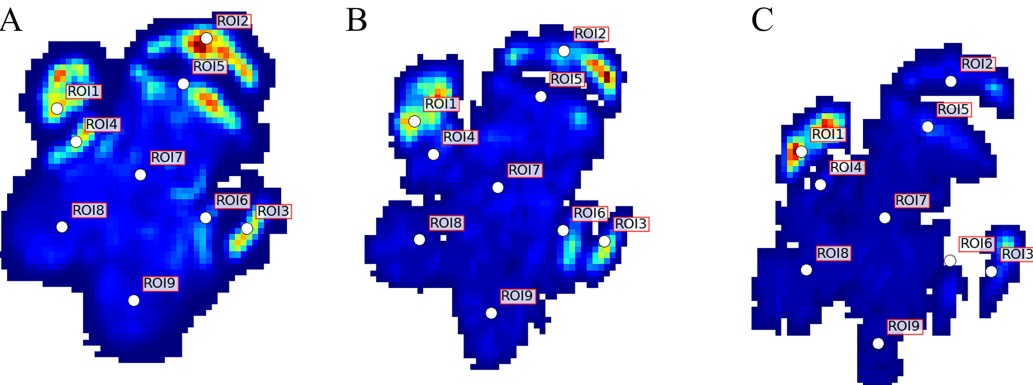

**Figure 6 Peak pressure patterns during the whole stance phase of the right hindfoot of subjects 1 (A and B) and 2 (C).**

hindfoot (Fig. 2). Such variations may be due to the ROI method used for data analysis. Although the ROI approach is a widely used technique for the estimation of peak pressure magnitudes sampled from specific anatomical regions, it overlooks variability within regions, assuming that all regions are functionally independent. We have previously shown a significant interaction between the topology of the ROIs and pressure magnitudes in elephants (*Panagiotopoulou et al., 2016*). Variation in peak pressures between ROIs may also have a biological importance considering that the left hindfoot sometimes showed a lateral-caudal-central roll off pattern, but we remain conservative with any biological conclusions due to our experimental and sample size limitations.

The general COP trajectories in our rhinoceros subjects were similar to elephants in being linear during the final half of stance phase rather than sigmoidal as in humans (*Lord, Reynolds & Hughes, 1986*) and bonobos (*Vereecke et al., 2003*). However, contrary to elephants, our rhinoceros subjects loaded the medial part of the foot at impact and then shifted their load centrally during mid-stance prior to toe-off via their central digit. Reasons for this apparent preference to avoid "heelstrike," and the strongly medially-biased COP pattern in our subjects early in stance phase, remain unclear. Nevertheless, the variability of COP patterns is cause for caution in attributing this pattern to all rhinoceroses until more such data can be obtained and compared. However, this medial bias early in stance phase does, tantalizingly, fit with the pattern observed by *Von Houwald (2001)* in Indian rhinoceroses, in which the medial angle of the foot tended to develop cracks and similar wear earlier than other regions. Hints at other unusual COP patterns—or perhaps subject variability or measurement error—in large mammals (e.g., hippopotamus and tapir COP traces in Fig. 1 of *Michilsens et al., 2009*) are further cause for caution and future analyses. More detailed studies of cows, for example (*Van Der Tol et al., 2002*, *2003*, *2004*) indicate further apparent interspecific variation, such as cows tending to have larger pressures on their lateral (vs. medial) horns (i.e., claws), and having their forefeet more

evenly loaded throughout stance phase vs. hindfeet with pressure magnitudes that shift from lateral at heelstrike to medial at toe-off.

Due to this variation in rhinoceroses' foot pressures and COP trajectories, locomotor patterns are important for assessing peak pressure distributions qualitatively. The peak pressure "heat maps" for all subjects and steps shown in Figs. 3–6 indicated a clear concentration of peak pressures around the horn and phalangeal pads of all three digits. These results tentatively support our hypothesis II—that peak pressures are evenly distributed, rather than biased toward the central and lateral digits, which corresponds to the relatively even distribution of osteopathologies across digits II–IV (*Regnault et al., 2013*). An even distribution of peak pressures across all three digits might be a by-product of the horizontal position of the foot at impact as manifested by the COP traces (i.e., avoidance of heelstrike). Regardless, large animals such as elephants and rhinoceroses clearly use enlarged foot contact areas to protect the digits from peak pressures that otherwise could cause tissue damage (*Chi & Roth, 2010*; *Michilsens et al., 2009*).

It is also interesting that forefoot pressures were normally higher in our three subjects, and forefoot osteopathologies tend to be more common than hindfoot osteopathologies (*Regnault et al., 2013*)—although one study found more chronic foot disease overall in the hindfeet, rather than forefeet, for a sample of one-horned rhinoceroses (*Von Houwald & Flach, 1998*). The latter study posited some biomechanical factors that may underlie foot pathologies, including toe horn-cracking, shearing forces on the middle toe, low friction causing low wear, and overgrowth of the middle toe horn, which could inspire future studies building on this one. Regardless, these patterns are opposite those tentatively identified for elephants sampled by *Regnault et al. (2017)*—they found no clear forefoot vs. hindfoot differences in osteopathologies despite some evidence for higher pressures on elephant forefeet (*Panagiotopoulou et al., 2012*, *2016*). It is tempting to speculate that the more similar morphology and presumably function of all four rhinoceros feet compared with the disparate morphology of elephant fore—feet vs. hindfeet may explain these discrepancies, but such speculations demand cautious future analysis.

Many factors account for osteopathologies such as congenital, developmental, metabolic, diet, age, traumatic injuries (summarised in *Galateanu et al., 2013*). However, captivity in enclosures with limited space for the animals to remain athletic, and exposure to hard concrete for long hours may exacerbate foot disease even if not the primary cause. To better understand foot pressures in rhinoceroses and the links to foot disease, more in vivo locomotor data are required; ideally from multiple species and management regimes. *Von Houwald (2001)* speculated that wild rhinoceroses walk on their soles (phalangeal pads) whereas captive rhinoceroses walk more on their fat pads. It would be fascinating to investigate this possibility using pressure pad analyses in the future.

Contrary to elephants that can easily be trained to walk over a walkway lined with pressure plates using food as encouragement, rhinoceroses are seldom well-trained, so in vivo data collection is challenging. We initially collected data on five animals but only a limited number of trials from this study's three individuals could be used for final analyses due to spatial (i.e., partial foot contacts) and temporal (i.e., starting data collection after initial foot contact and/or terminating data collection before final foot contact)
completeness issues. We conducted a power analysis for a one-way ANOVA on our rhinoceros peak pressure data for each foot, where omega-squared was used for the effect size, significance was set at 0.01 and power was set at 0.8. The minimum number of rhinoceroses to achieve this power would be 8, 39, 29, and 13 for the left forefoot, right forefoot, left hindfoot, and right hindfoot datasets, respectively. Considering accessibility and experimental limitations, it will be difficult (if not impossible) to recruit enough rhinoceroses (>40 considering that some subjects would need to be discarded from any study) from the same captive setting for a statistically robust experiment.

Habitat loss and poaching have brought many rhinoceros species, in particular the Javan and the Sumatran, to the brink of extinction (*Crosta, Sutherland & Talerico, 2017*). Despite on-going legal and conservation efforts to protect rhinoceroses, the number of populations impacted by poaching has increased dramatically over the last two decades, with South Africa being affected the most due to having the largest number of rhinoceroses in the world (*Charlton, 2017*; *Crosta, Sutherland & Talerico, 2017*). One of the measures in place to protect these animals from extinction is to keep and breed them in captivity. While in captivity, they may develop foot disorders, in particular chronic foot diseases, osteoarthritis, bone remodelling, osteitis-osteomyelitis, pododermatitis, abscesses, and fractures (*Galateanu et al., 2013*; *Jacobsen, 2002*; *Regnault et al., 2013*; *Von Houwald & Flach, 1998*) that compromise animal welfare or even cause mortality due to being painful, progressive and often untreatable. Even in wild rhinoceroses, there are reports of serious foot disease (*Zainuddin et al., 1990*), and a high incidence of osteopathology appears to be an ancestral evolutionary trait for the lineage, which may complicate efforts to improve the welfare of rhinoceroses (*Stilson, Hopkins & Davis, 2016*). To date, most focus on appendicular pathologies in extant rhinoceroses have been on the feet, but the latter study's finding that pathologies have been equally prevalent across the limbs throughout the ~50 million year history of Rhinocerotidae raises questions of whether more proximal limb pathologies remain common but overlooked in captive rhinoceroses. Follow-up studies should investigate this question and even integrate it with biomechanical analyses to test whether some regional mechanical stresses are unusually high and corresponding with locations predisposed to pathologies. *Alexander & Pond (1992)* used a very simple analysis to estimate that femur safety factors were high in a galloping White rhinoceros but this method certainly is imprecise, and stresses in the humerus or zeugopodial elements are unknown— as are any joint contact stresses, which should be more important for pathologies.

Disease management in large mammals such as elephants and rhinoceroses can be challenging and examination using diagnostic approaches requires general anaesthesia or sedation, which can have negative effects on the animal (*Gage, 2006*; *Hittmair & Vielgrader, 2000*; *Siegal-Willott, Alexander & Isaza, 2012*; *Von Houwald & Flach, 1998*). These challenges, coupled with the fact that foot diseases may only clearly manifest themselves when they have progressed to advanced stages, can make euthanasia an unavoidable outcome (*Jones, 1979*; *Mikota, 1999*; *Mikota, Sargent & Ranglack, 1994*). The causes of foot pathologies are multifactorial (*Csuti, Sargent & Bechert, 2001*), but the biomechanical pressures imposed during locomotion presumably can promote or worsen them. How

can we thus protect rhinoceroses from developing foot diseases, or monitor treatment vs. progression of chronic foot disease? An important step is to learn how rhinoceros feet function in captivity. A valuable follow-on step would be to examine how husbandry conditions in captivity affect innate foot function. Nevertheless, whilst we have a fair understanding of elephant foot pressures from captive and semi-wild settings (*Panagiotopoulou et al., 2012*, *2016*), here, we have added new data on the pressures that White rhinoceroses routinely apply to their feet during normal locomotion. Our foot pressure data give tentative insights into not only basic biomechanics but also potential links of normal form and function vs. mechanically-induced foot disease.

## CONCLUSIONS

We conclude that there is tentative support for our hypothesis I, that peak locomotor pressures during walking in White rhinoceroses are the lowest in the central and caudal parts of the foot at the locations of the fat pad and its lobes, as in elephants. We also found support for our hypothesis II, that peak pressures are equally concentrated around the horns/hooves and phalangeal pads of digits II–IV (unlike elephants) instead of being concentrated more laterally onto digits III and IV (analogous to elephants). This finding concurs with the incidence of osteopathologies, bolstering the proposition that there is a link between locomotor pressures during walking and such pathologies (*Regnault et al., 2013*, *2017*).

## ACKNOWLEDGEMENTS

We thank the keepers and members of staff at the Colchester Zoo, UK for their assistance with the experiments. We also thank RVC graduates Katherine Jones, Richard Harvey, and Keri Holmes for assistance with data collection. Thanks are due to Hyab Mehari Abraha from Monash University and the Monash Bioinformatics Platform for technical support. We are also grateful for the two peer reviewers' constructive critiques.

### Funding

This project was supported by Biotechnology and Biological Sciences Research Council (UK) grant number BB/H002782/1 to John R Hutchinson. The funders had no role in study design, data collection and analysis, decision to publish, or preparation of the manuscript.

### Grant Disclosure

The following grant information was disclosed by the authors:
Biotechnology and Biological Sciences Research Council: BB/H002782/1.

### Competing Interests

John R. Hutchinson is an Academic Editor for PeerJ.

## Author Contributions

- Olga Panagiotopoulou conceived and designed the experiments, performed the experiments, analyzed the data, prepared figures and/or tables, authored or reviewed drafts of the paper, approved the final draft.
- Todd C. Pataky analyzed the data, contributed reagents/materials/analysis tools, prepared figures and/or tables, authored or reviewed drafts of the paper, approved the final draft.
- John R. Hutchinson conceived and designed the experiments, performed the experiments, authored or reviewed drafts of the paper, approved the final draft.

## Animal Ethics

The following information was supplied relating to ethical approvals (i.e., approving body and any reference numbers):

The Royal Veterinary College's Animal Ethics Committee (Approval number URN 2010 1052).

## Data Availability

Hutchinson, John; Panagiotopoulou, Olga (2019): Experimental raw pressure pad data from White rhinoceroses. figshare. Dataset. https://doi.org/10.6084/m9.figshare.7608797.v1

## Supplemental Information

Supplemental information for this article can be found online at http://dx.doi.org/10.7717/peerj.6881#supplemental-information.

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
