# Peer review of "Foot pressure distribution in White Rhinoceroses (Ceratotherium simum) during walking"

_PeerJ, doi:10.7717/peerj.6881_

## Round 0.1 · original submission · Major Revisions

Both reviewers have commented positively that your study is well-presented and will be a valuable addition to the literature. I have received some minor comments and some more major suggestions, and deem your revision to fall somewhere in between. Reviewer #1 has suggested that the ms would benefit from a more detailed comparison between rhinos and elephants, in particular, and the use of statistical analyses (while recognizing their limitations herein). Reviewer #2 has provided some suggestions to improve Figure 1. In addition, please respond to the request for provision of additional raw data (code) as this would enhance the value of your study for the research community. I look forward to receiving your revision.

·

Basic reporting

no comment

Experimental design

see below

Validity of the findings

see below

Additional comments

This paper examines foot pressure distribution in white Rhinos with a focus on the relative pressure produced under the footpad
Overall I think this paper presents valuable data which would be of interest to biomechanists studying foot pressure distributions.
However, there appears to be some information missing from the methods, and a lack of statistical testing of the results.
I have added some detailed comments below.

Intro
The intro seems to state that this is an exploratory study, which is fine, but it does include hypotheses which are tested. Which seems to contract the intro.
I think the authors might also comment further on the comparison between elephants and rhinos. Infact I think the study could be highly hypothesis driven if the question compares the foot pressure distributions on 5 toed elephants with three toed rhinos. Does the reduced number of toes increase peak pressures?

Methods
The methods seems to be missing a section describing how many trials were collected from each individual and foot.
From the figures we can see that a large number of trials were included - but i could not find where the exact sample size was reported.

Results
Given that there appears to be a moderately large sample size, it seems strange that no statistical analysis is attempted. The results report which pressures were the highest etc, but no indication if these were significantly higher.
Even if the statistical results are not significant, I think its still important to report them.
I suggest a linear mixed effect analysis, or ANOVA, include both ROI and foot (and speed?), with subject as a random effect.

Finally i think a better comparison could be made between elephants and rhinos. Perhaps you could allow for the different force plates used in each study by comparing the relative peak pressures to
those of the central ROI (which i think is usually the lowest?). I.e. how many percent higher is the peak ROI in elephant, compared with its central ROI, in comparison with the rhino? Does the reduction in toe number result in higher relative stresses around these digits?

Reviewer 2 ·

Basic reporting

The basic reporting is good. In a few cases sentences become too long for clarity (e.g., the second sentence of the introduction that begins with "Extant rhinoceroses" and spans too many ideas for a single sentence). In some places semicolons are incorrectly used in place of a comma: see lines 56, 80, 94, 197, and 228. There should be an additional semiocolon after the closed parenthetical on line 131 to complete the list. Outside of these small corrections, the language is clear throughout. The literature appears to be suitably addressed. It would be interesting to know about the evolutionary history of foot pathology in other very large terrestrial mammals (e.g. the extinct brontotheres and chalicotheres, or other extinct proboscideans), but that curiosity is obviously outside the scope of this study and the authors need not address it (I am unaware of any existing research to cite).

In Figure 1B, the labels for regions of interest are too small to be clear, even very zoomed-in. The dots should be made clearer (even if larger is not possible, they should be outlined in a brighter, contrasting color). I suggest the labels for each dot be shown outside of the foot area, with a line connecting each label box to the appropriate dot on the pressure diagram. Alternatively, separate this into a pressure diagram (as an example of data) and a separate line drawing of the feet (which can more clearly show each ROI location).

Raw data pressure data is not provided as far as I can tell, nor is code to extract ROI or COP from said data. Images of pressure and COP trajectories are provided (though these might be easier to interpret if also made into supplemental videos), as well as peak pressures, walking speed, etc at each ROI for the images provided. The data included are sufficient to evaluate the descriptions and conclusions of the paper, so I leave it to the editor whether this is sufficient sharing of raw data.

The article is a self-contained and useful contribution to the literature, with a clearly defined scope and relevant results.

Experimental design

The research question is well-defined and meaningful. The methods are clear and sufficiently described. I have no suggested changes here.

Validity of the findings

The authors clearly recognize the statistical limitations of this study due to small sample size, and they do not overstep the limits of a qualitative interpretation. Speculation is clearly noted as such, and I appreciate the power analysis indicating the number of specimens that would be necessary for certain analyses.

Additional comments

Is there any research comparing osteopathology in captive vs. wild rhinoceros? The authors do note that the evolutionary history of pathology in rhinoceros feet complicates simple husbandry questions such as whether it is even possible to prevent disease, but if such a reference exists it would be valuable to include (or to point out that, if it does not exist, it would be a valuable contribution to this exploration of rhinoceros foot health).

This is an interesting study and I look forward to seeing it published.

---

## Round 0.2 · Minor Revisions

Thank you for addressing the previous round of reviewer comments. Upon re-review, reviewer #2 is happy with the changes that you've made to the manuscript and the comments in response to the earlier suggestions. As such, they suggest the paper is ready for acceptance. I have re-reviewed it myself and I agree with Reviewer #2, this is a well-written and carefully prepared paper and the challenges associated with collecting these data (and thus limits to sample sizes) are clearly outlined.

I have a couple of very minor points:

ln100 and 105: please insert spaces between numbers and meters, i.e. 0.4 m
ln116: typo, should be ''x, y, time''
I think Figure 2 would benefit from changing the ellipse colours to be more contrasting - the greyscale is quite difficult to appraise. Alternatively you could use filled/open ellipses and an additional shape.

Reviewer 2 ·

Basic reporting

no comment

Experimental design

no comment

Validity of the findings

no comment

Additional comments

In my opinion, the authors have fully addressed the concerns from the previous round of review. I am inclined to agree with their comments about statistical hypothesis-testing and whether it is appropriate to conduct formal tests on their data, given the sample size. I think this study has clear value as an exploratory analysis, and look forward to seeing it published.

---

## Round 0.3 · accepted · Accept

Thank you for making the two, minor changes related to formatting that were suggested in the last round of review. These were very minor comments and I'm happy to recommend your paper for publication. I look forward to seeing your paper published.

#